# On Transfer of Adversarial Robustness from Pretraining to Downstream Tasks

**Laura F. Nern**
Yahoo Research
laurafee.nern@yahooinc.com

**Harsh Raj**
Delhi Technological University
harsh777111raj@gmail.com

**Maurice Georgi**
Hyundai Mobis
georgimaurice@gmail.com

**Yash Sharma**
University of Tübingen
yash.sharma@bethgelab.org

## Abstract

As large-scale training regimes have gained popularity, the use of pretrained models for downstream tasks has become common practice in machine learning. While pretraining has been shown to enhance the performance of models in practice, the transfer of robustness properties from pretraining to downstream tasks remains poorly understood. In this study, we demonstrate that the robustness of a linear predictor on downstream tasks can be constrained by the robustness of its underlying representation, regardless of the protocol used for pretraining. We prove (i) a bound on the loss that holds independent of any downstream task, as well as (ii) a criterion for robust classification in particular. We validate our theoretical results in practical applications, show how our results can be used for calibrating expectations of downstream robustness, and when our results are useful for optimal transfer learning. Taken together, our results offer an initial step towards characterizing the requirements of the representation function for reliable post-adaptation performance.[1]

## 1 Introduction

Recently, there has been a rise of models (BERT [31], GPT-3 [7], and CLIP [43]) trained on large-scale datasets which have found application to a variety of downstream tasks [6]. Such pretrained representations enable training a predictor, e.g., a linear "head", for substantially increased performance in comparison to training from scratch [10, 11], particularly when training data is scarce.

Generalizing to circumstances unseen in the training distribution is significant for real-world applications [2, 60, 29], and is crucial for the larger goal of robustness [27], where the aim is to build systems that can handle extreme, unusual, or adversarial situations. In this context, adversarial attacks [51, 5], i.e., inducing model failure via minimal perturbations to the input, provide a stress test for predictors, and have remained an evaluation framework where models trained on large-scale data have consistently struggled [53, 21, 22], particularly when, in training, adversarials are not directly accounted for [23, 40].

With that said, theoretical evidence [8] has been put forth that posits overparameterization is required for adversarial robustness, which is notable given its ubiquity in training on large-scale datasets. Thus, given the costs associated with large-scale deep learning, we are motivated to study how adversarial robustness transfers from pretraining to downstream tasks.

---

[1] https://github.com/lf-tcho/robustness_transfer

37th Conference on Neural Information Processing Systems (NeurIPS 2023).

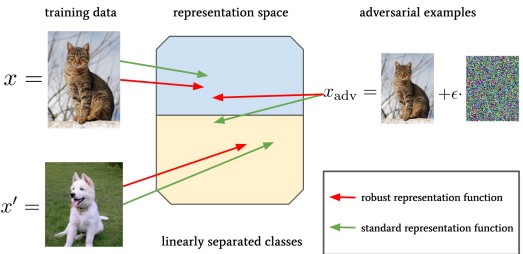

Figure 1: **Overview**. We distinguish between a robust (red) and non-robust (green) representation by its sensitivity to adversarial attacks, and use both for linear separation of the representation space, e.g., binary classification. This illustrates the dependence of the classifier on the representation's robustness, i.e., the predictor can only be robust if $x_{adv}$ and $x$ can be linearly separated from $x'$ in representation space.

For this, considering the representation function to be the penultimate layer of the downstream predictor, we contribute theoretical results which bound the robustness of the downstream predictor by the robustness of the representation function. Fig. 1 illustrates the intuition behind our theory, and shows that a linear classifier's robustness is dependent on the robustness of the pretrained representation. Our theoretical results presented in Sec. 3 and Sec. 4 are empirically studied in Sec. 5, where we validate the results in practice, and in Sec. 6 to show how the theory can be used for calibrating expectations of the downstream robustness in a self-supervised manner [4]. In all, our theoretical results show how the quality of the underlying representation function affects robustness downstream, providing a lens on what can be expected upon deployment. We present the following contributions:

- we theoretically study the relationship between the robustness of the representation function, referred to as *adversarial sensitivity (AS) score*, and the prediction head. In Thm. 1, we prove a bound on the difference between clean and adversarial loss in terms of the AS score. In Thm. 2, we prove a sufficient condition for robust classification again conditioned on the robustness of the representation function.

- we evaluate the applicability and consistency of our theory in applications in Sec. 5.1 and Sec. 5.2. Furthermore, in Sec. 6, we illustrate the utility of the AS score to predict expected robustness transfer via linear probing.

- in Sec. 6.2, we study how linear probing compares to alternatives, like full finetuning, in robustness transfer for robust performance downstream.

## 2 Related Work

**Transfer Learning:** While empirical analyses of pretraining abound [33, 12, 62, 42, 3, 48], a number of theoretical analyses have also been contributed [56, 52, 19, 14, 49, 35, 38]. The majority of the theoretical studies aim to understand the benefit of transfer learning with respect to improved performance and data requirements compared to training from scratch. Therefore, the referenced studies include results on sample complexity [52, 19, 14, 49] and generalization bounds [38] for fine-tuning and linear probing in various scenarios. In this work, we primarily focus on the linear probing approach and perform a theoretical analysis to investigate its efficiency in transferring adversarial robustness.

**Robustness:** There is a direct connection between the mathematical formulation of adversarial robustness and the definition of a local Lipschitz constant of the network [25], but computing this quantity is intractable for general neural networks. However, there exists a line of work providing more practical approaches in certifiable robustness [54, 44, 15, 37, 36, 46], which can provide guarantees on the detection of adversarial examples. Relative to empirical approaches [23, 40, 18, 48, 55], certifiable methods have comparatively struggled to scale to datasets such as ImageNet [17]. Furthermore, likely due to computational expense, adversarial training has not yet been adopted by models trained on large-scale datasets [31, 7, 43]. Still, a number of works have incorporated adversarial training into

pretraining, in particular, self-supervised contrastive learning [32, 30, 9, 20]. With that said, our work is focused on the analysis of robustness irrespective of the pretraining approach.

Regarding analysis, [59] studied the trade-off between accuracy and robustness, and showed that, under a set of separation conditions on the data, a robust classifier is guaranteed to exist. Their work focuses on proving the robustness of a network itself and therefore requires Lipschitzness of the model. Further, [8] showed that overparameterization is needed for learning such smooth (i.e. Lipschitz) functions, which is critical for robustness. On the other hand, we specifically study the relationship between the robustness of the representation function and the prediction head, a scenario consistent with the practice of pretraining for transfer learning.

**Transfer Learning & Robustness:** The question of how robustness is transferred from pretraining to downstream task has been studied empirically in the prior literature [50, 58, 28, 63]. Notably, while [50] observe that robust representations underly robust predictors, [63] also observe that non-robust representations underly non-robust predictors. An empirical analysis of adversarial robustness transfer for the special case of linear probing is included in ,e.g., [50]. Our work not only derives theoretical results and concrete proof for the empirical phenomena reported in these studies but also provides further empirical investigation.

## 3 Robustness Bound

Our first theoretical result concerns the discrepancy between the standard and adversarial loss of a neural network $f$. The considered loss function $\ell$ is evaluated on a dataset of input samples $\mathbb{X} = \{x_1, \ldots, x_n\} \subset \mathcal{X} \subset \mathbb{R}^d$ and labels $\mathbb{Y} = \{y_1, \ldots, y_n\}$, where $x_i$ is sampled from $\mathcal{X}$ with probability distribution $\mathcal{P}(\mathcal{X})$.

For the network, we consider a model $f : \mathbb{R}^d \to \mathbb{R}^c$ with a final linear layer, i.e. $f(x) = f_{W,\theta}(x) = W g_\theta(x)$. The function $g_\theta : \mathbb{R}^d \to \mathbb{R}^r$ can be considered as a representation function for the input parameterized by $\theta$, and $W \in \mathbb{R}^{c \times r}$ is the weight matrix of the linear layer. Note that this composition is both standard practice [10, 11], and has been extensively studied [56, 52, 19] in the literature.

For a given data pair $(x_i, y_i)$, we are interested in the stability of the model's prediction of $y_i$ with respect to (w.r.t.) perturbations of the input $x_i$ as our robustness measure. Specifically, for $\delta \in \mathbb{R}^d$, we find the maximum loss $\ell(f(x_i + \delta), y)$ under a specified constraint $\|\delta\| < \epsilon$ based on some norm $\|\cdot\|$. Note that the defined measure is commonplace in the literature on adversarial attacks [51, 23].

We introduce the following notation for the standard loss $\mathcal{L}(f_{W,\theta})$ and the adversarial loss $\mathcal{L}_{\mathrm{adv}}(f_{W,\theta})$,

$$\mathcal{L}(f_{W,\theta}) := \mathbb{E}_{\mathcal{P}(\mathcal{X})}[\ell(f_{W,\theta}(x), y)],$$

$$\mathcal{L}_{\mathrm{adv}}(f_{W,\theta}) := \mathbb{E}_{\mathcal{P}(\mathcal{X})}\left[\max_{\|\delta\| < \epsilon} \ell(f_{W,\theta}(x + \delta), y)\right].$$

We aim to bound the effect of adversarial attacks in terms of their impact on the representation function, i.e. $\|g_\theta(x_i) - g_\theta(x_i + \delta)\|_2$. To obtain such a result for a general loss function $\ell$, the variation in the loss must be bounded in relation to the proportional change in $f(x_i)$. This condition is formalized by the concept of Lipschitzness,

**Definition 1.** *A function $\ell : \mathbb{R}^d \to \mathbb{R}$ is C-Lipschitz in the norm $\|\cdot\|_\alpha$ on $\mathcal{R} \subset \mathbb{R}^d$, if*

$$|\ell(r) - \ell(r')| < C \|r - r'\|_\alpha,$$

*for any $r, r' \in \mathcal{R}$.*

The following theorem is valid for any loss function $\ell$ that satisfies Defn. 1 w.r.t. $L_1$, $L_2$ or $L_\infty$ norm.

**Theorem 1.** *We assume that the loss $\ell(r, y)$ is $C_1$-Lipschitz in $\|\cdot\|_\alpha$, for $\alpha \in \{1, 2, \infty\}$, for $r \in f(\mathcal{X})$ with $C_1 > 0$ and bounded by $C_2 > 0$, i.e., $0 \le \ell(r, y) \le C_2 \ \forall r \in f(\mathcal{X})$. Then, for a subset $\mathbb{X}$ independently drawn from $\mathcal{P}(\mathcal{X})$, the following holds with probability of at least $1 - \rho$,*

$$\mathcal{L}_{\mathrm{adv}}(f_{W,\theta}) - \mathcal{L}(f_{W,\theta}) \ \le \ L_\alpha(W) \, C_1 \, \frac{1}{n} \sum_{i=1}^n \max_{\|\delta\| < \epsilon} \|g_\theta(x_i + \delta) - g_\theta(x_i)\|_2 \ + \ C_2 \sqrt{\frac{\log(\rho/2)}{-2n}},$$

*where*

$$L_\alpha(W) := \begin{cases} \|W\|_2 & , \text{if } \|\cdot\|_\alpha = \|\cdot\|_2, \\ \sum_i \|W_i\|_2 & , \text{if } \|\cdot\|_\alpha = \|\cdot\|_1, \\ \max_i \|W_i\|_2 & , \text{if } \|\cdot\|_\alpha = \|\cdot\|_\infty. \end{cases}$$

The proof of the theorem can be found in Appx. A.1. Note that the norm used to define the perturbation $\delta$ is arbitrary and only needs to be consistent on both sides of the bound.

**Proof sketch:** First, we apply Hoeffding's inequality, which requires a fixed function $f_{W,\theta}$, independently drawn samples $\{x_1, \ldots, x_n\}$, and an upper bound $C_2$ on the loss function. Next, we bound the average losses utilizing the $C_1$-Lipschitzness in $\|\cdot\|_\alpha$ of the loss function. Further, the Cauchy-Schwarz inequality is used to seperate the norms of $W$ and the representation function $g_\theta$.

**Intuition:** Thm. 1 shows that the effect an $\epsilon$-perturbation has on the representation function $g_\theta$ indeed upper bounds the difference between the standard and adversarial loss. Thus, we characterize the relationship between how vulnerable the linear predictor and the underlying representation is to adversarial examples. Further, Thm. 1 formalizes the impact the weight matrix $W$ has on robustness transfer and thereby provides a theoretical argument for regularization in this context.

The statement of Thm. 1 is formulated in terms of the expectation of the losses over $\mathcal{P}(\mathcal{X})$. In particular, the second summand on the right hand-side (RHS) of the bound illustrates the finite-sample approximation error between the expectation and the average over $n$ samples in $\mathbb{X}$. The following lemma provides a version of the theorem for the average difference of the losses over a fixed finite sample $\mathbb{X}$, which can directly be concluded from the proof of Thm. 1 in Appx. A.1. We use this lemma to study our theoretic result in the experiments in Sec. 5.1.

**Lemma 1** (Finite-sample version of Thm. 1). *We assume that the loss $\ell(r, y)$ is $C$-Lipschitz in $\|\cdot\|_\alpha$, for $\alpha \in \{1, 2, \infty\}$, for $r \in f(\mathcal{X})$ with $C > 0$. Then, the following holds for any dataset $(\mathbb{X}, \mathbb{Y})$,*

$$\frac{\frac{1}{n}\sum_{i=1}^n \max_{\|\delta\|<\epsilon} \ell(f_{W,\theta}(x_i+\delta), y_i) - \ell(f_{W,\theta}(x_i), y_i)}{L_\alpha(W)\,C} \leq \frac{1}{n}\sum_{i=1}^n \max_{\|\delta\|<\epsilon} \|g_\theta(x_i+\delta) - g_\theta(x_i)\|_2.$$

Lemma 1 shows that the normalized difference between average clean and adversarial loss can be bound by the average robustness of the representation function on the RHS, which we denote as the representation function's *adversarial sensitivity (AS) score* on the dataset $\mathbb{X}$.

## 3.1 Valid Loss Functions

To elaborate on the applicability of the result, we discuss exemplary scenarios in which the assumptions of our theory are fulfilled in practice. As a first example, we consider a classification task where we use the softmax cross-entropy loss. While the cross-entropy loss itself is not Lipschitz continuous, since the logarithm is not, the softmax cross-entropy loss is indeed Lipschitz. We define the softmax cross-entropy loss as follows,

$$\ell_{CE}(f(x), y) = -\log(p_y(x)) = -\log\left(\frac{\exp\{f(x)_y\}}{\sum_{i=1}^c \exp\{f(x)_i\}}\right),$$

where $p_y(x)$ denotes the y-th entry of the softmax function applied to the final layer, $f(x)_y$ is the y-th entry of the model output with $f(x)_y = W_y g(x)$ and $W_y$ being the y-th row of the matrix $W$. In Lemma 3 (in Appx. A), we prove that $\ell_{CE}$ is 2-Lipschitz in $\|\cdot\|_\infty$.

Further, the theory is also valid for regression tasks, where the mean square error (MSE)

$$\ell_{MSE}(f(x), y) = \|f(x) - y\|_2$$

is used to measure the loss. This loss function is 1-Lipschitz in $\|\cdot\|_2$, since, by the reverse triangle property, it holds that

$$|\|f(x_1) - y\|_2 - \|f(x_2) - y\|_2| \leq \|f(x_1) - f(x_2)\|_2.$$

Additional losses which satisfy Defn. 1 include the logistic, hinge, Huber and quantile loss [13].

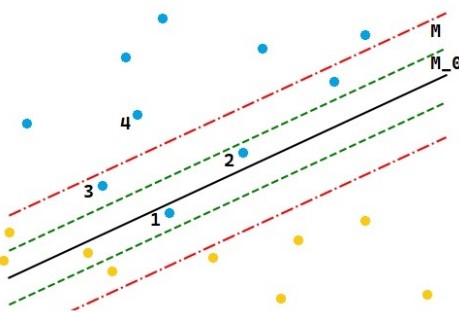

Figure 2: The graphic shows two classes of datapoints and a linear classifier's decision boundary (black). The line $M_0$ (green) is the optimal margin defining the boundary between a successful or failed adversarial attack. Line $M$ (red) represents the margin derived from Thm. 2 which is an upper bound on $M_0$. Four datapoints from the blue class are highlighted: 1 is wrongly classified, 2 is a non-robust example, 3 is robust but does not fulfill equation (1), and 4 is a robust classification fulfilling the bound in (1).

## 4  Robustness Classification Criterion

In Sec. 3, we presented a theoretical result independent of any particular downstream task and valid for various loss functions. Here, we focus on classification tasks, and, in this context, we provide an additional statement that studies the robustness of a linear classifier w.r.t. adversarial attacks. We refer to a classifier as robust if there does not exist a bounded input perturbation that changes the classifier's prediction.

In particular, we consider the classifier

$$c(x) = \mathrm{argmax}_i \, f(x)_i,$$

where $f(x) = W g_\theta(x)$ is the model as defined in Sec. 3. The following theorem provides insights on how the robustness of the underlying representation function $g_\theta$ impacts the robustness of a linear classifier. Specifically, it formally describes how the robustness of the representation, i.e. the AS score on a datapoint $x$, determines the set of weight matrices which yield a robust classifier.

**Theorem 2.** *The classifier $c(x)$ is robust in $x$, i.e., $c(x + \delta) = c(x)$ for $\delta \in \mathbb{R}^d$ with $\|\delta\| < \epsilon$, if for $x \in \mathcal{X}$ with $c(x) = y$ it holds that*

$$\|g_\theta(x+\delta) - g_\theta(x)\|_2 \;\leq\; \min_{j \neq y} \frac{|f(x)_y - f(x)_j|}{\|W_y - W_j\|_2}. \tag{1}$$

The proof of the theorem can be found in Appx. A.2.

**Proof sketch:** To prove robustness, we need to show that $f(x + \delta)_y > f(x + \delta)_j$ if $f(x)_y > f(x)_j$. The proof consists of rewriting this condition and applying the Cauchy-Schwarz inequality to separately bound in terms of the difference in $W$ and $g_\theta$.

**Intuition:** Thm. 2 shows that the effect an $\epsilon$-perturbation has on the representation function $g_\theta$ determines the necessary class separation by the linear predictor for robust classification to be ensured. Specifically, the RHS of Thm. 2 can be understood as a normalized margin between class logits for $x$, where the classification margin refers to the difference between the classification score for the true class and maximal classification score from all false classes. Thus, the theorem states that if the margin is larger than the effect of an $\epsilon$-perturbation on the representation function, then the classifier is robust. Note that if a datapoint does not satisfy the bound, i.e. LHS is greater than the RHS, the robustness of the classifier is ambiguous. In this case, the RHS margin is not the exact separation, or the optimal margin, between robust and non-robust classification. Fig. 2 provides intuition for this interpretation of the result in a binary classification scenario.

# 5 Empirical Study of Theory

The following experiments serve to validate the theoretical results in practical applications. For the finite sample version of Thm. 1, Lemma 1, we consider a classification task in Sec. 5.1, and a regression task in Appx. B.2. In Sec. 5.2, we study the robustness condition established in Thm. 2.

In the following sections, we train a linear predictor on top of a pretrained representation function for a specific downstream task, i.e., we perform transfer learning via linear probing. To study robustness transfer, we use robustly pretrained representation functions and examine if the transferred robustness is consistent with the theoretical bounds. As our robust representation function, we use models robustly pretrained on CIFAR-100 [1, 34], and ImageNet [47, 17]. Both models are available on RobustBench [16] and are adversarially trained using the softmax cross-entropy loss and $L_\infty$-attacks. For our downstream tasks, we consider CIFAR-10 [34], Fashion-MNIST [57] and Intel Image[2]. Further experimental details can be found in Appx. B.1.

## 5.1 Classification - Cross-Entropy Loss

Here, we empirically evaluate Lemma 1, the finite sample version of Thm. 1, on classification tasks. As per Sec. 3.1, for softmax cross-entropy, we can apply Lemma 1 with Lipschitz constant $C = 2$ and mapping function $L_\infty(W)$.

In Tab. 1, we present the left-hand side (LHS), the scaled difference between clean and robust cross-entropy (CE) loss, and the right-hand side (RHS), the representation function's AS score, of Lemma 1. Note that, for LHS, the attack on $f$ maximizes the CE loss, while for RHS, the attack on $g_\theta$ maximizes the impact on the representation function, i.e. $\|g_\theta(x_i) - g_\theta(x_i + \delta)\|_2$. See Appx. B.1 for further details.

The results in Tab. 1 validate our theoretical predictions in practice, as the LHS is indeed always upper bounded by the RHS. Furthermore, we see signs of correlation between the LHS and RHS, i.e. significant changes on one side of the bound are accompanied by similar changes on the other. Tab. 2 further highlights the connection between our result and weight norm regularization, as it confirms that a smaller weight norm, i.e., smaller $L_\alpha(W)$, corresponds to a smaller decrease in robustness, i.e., a smaller difference between the losses.

Table 1: For the CIFAR-100 and ImageNet pretraining datasets, the LHS and RHS, i.e., the AS score, of Lemma 1 are computed for each of the CIFAR-10, Fashion-MNIST and Intel Image downstream tasks. Additionally, $L_\infty(W)$ and the absolute difference between clean and adversarial CE loss are provided.

|  | CIFAR-100 | | | ImageNet | | |
|---|---|---|---|---|---|---|
|  | CIFAR-10 | F-MNIST | Intel Image | CIFAR-10 | F-MNIST | Intel Image |
| Diff. CE | 0.74 | 0.8 | 0.47 | 4.14 | 9.4 | 0.14 |
| $L_\infty(W)$ | 4.4 | 4.6 | 3.1 | 1.9 | 2.0 | 1.4 |
| LHS | 0.08 | 0.09 | 0.08 | 1.11 | 2.33 | 0.05 |
| AS score | 0.98 | 2.11 | 1.05 | 15.7 | 53.2 | 2.75 |

Table 2: The same datasets and metrics are considered as in Tab. 1. Here, however, L2 weight matrix regularization ($\lambda = 0.01$) is employed for all downstream tasks.

|  | CIFAR-100 | | | ImageNet | | |
|---|---|---|---|---|---|---|
|  | CIFAR-10 | F-MNIST | Intel Image | CIFAR-10 | F-MNIST | Intel Image |
| Diff. CE | 0.46 | 0.51 | 0.38 | 3.57 | 7.3 | 0.14 |
| $L_\infty(W)$ | 2.1 | 2.3 | 2.2 | 1.4 | 1.5 | 1.2 |
| LHS | 0.11 | 0.11 | 0.09 | 0.81 | 2.43 | 0.06 |
| AS score | 0.98 | 2.11 | 1.05 | 15.7 | 53.2 | 2.75 |

---

[2]https://www.kaggle.com/datasets/puneet6060/intel-image-classification

## 5.2 Classification Criterion

Here, we focus on empirically analysing Thm. 2, which specifies an upper bound on the representation's sensitivity against $\epsilon$-perturbations, where the condition ensures robustness of the classifier against local perturbations for each datapoint.

In Tab. 3, we observe that indeed: if condition (1) is fulfilled, the classification is always robust, as can be seen from "rob. fulfilled". With that said, we do observe instances of robust classification which do not fulfill Eq. (1), which we proved ensures robustness, as can be seen from comparing "robust accuracy" to "prop. fulfilled". As with Lemma 1, we see that tighter results would be necessary for identifying all transferred robustness. Furthermore, as discussed in Sec. 4, since datapoints fulfilling our bound are even further away from the classification boundary than necessary, we also find that fulfilling the bound is predictive of robustness to stronger attacks, as can be seen in Appx. B.3.

For the scenario of pretraining on ImageNet and transferring to CIFAR-10 or Fashion-MNIST, almost no datapoint from the test set fulfills condition (1). This observation is accompanied by a steep drop from clean accuracy to robust accuracy, which suggests this is likely a consequence of the large distribution shift between ImageNet and the lower-resolution images of CIFAR-10 ($32 \times 32$) and Fashion-MNIST ($28 \times 28$) resized to ImageNet resolution ($256 \times 256$). Note that this distribution shift is evident regardless of how the images are scaled before applying the ImageNet representation function, see Appx. B.5. In Tab. 1, the correlation between distribution shift and loss of robustness is also captured prior to transfer learning by a large AS score, and Appx. B.5 provides further insight into this connection.

Table 3: For the CIFAR-100 and ImageNet pretraining datasets, we report clean & robust accuracy, the proportion of images fulfilling the theoretical bound, and, of the images fulfilling the theoretical bound, the proportion of images that are robust for each of the CIFAR-10, Fashion-MNIST and Intel Image downstream tasks.

|  | CIFAR-100 | | | ImageNet | | |
|---|---|---|---|---|---|---|
|  | CIFAR-10 | F-MNIST | Intel Image | CIFAR-10 | F-MNIST | Intel Image |
| clean accuracy | 70.9% | 84% | 78.5% | 92.4% | 87.2% | 91.2% |
| robust accuracy | 42.2% | 56.2% | 56.8% | 15.8% | 0% | 85.3% |
| prop. fulfilled | 13.4% | 10.1% | 26.3% | 0.1% | 0% | 56.2% |
| rob. fulfilled | 100% | 100% | 100% | 100% | - | 100% |

# 6 Adversarial Sensitivity Score

In the prior sections, we empirically validated that our theoretical results and required assumptions are realistic and hold in applications. Note that, in both of our results, the robustness of the representation function determines the robustness that transfers to downstream tasks. Specifically, Lemma 1 is stated using the *adversarial sensitivity (AS) score* of the representation function on a dataset $\mathbb{X}$,

$$\frac{1}{n} \sum_{i=1}^{n} \max_{\|\delta\| < \epsilon} \|g_\theta(x_i + \delta) - g_\theta(x_i)\|_2, \tag{2}$$

to bound the difference between clean and adversarial loss. Before specifying a downstream task, we can compute the representation function's adversarial sensitivity score on a set of unlabeled input images. In the following, we explore how to calibrate expectations on the robustness upon transfer of the representation function via linear probing to a particular downstream task based on the AS score.

## 6.1 Robustness Transfer via Linear Probing

Here, we explore how the AS score can be used to provide valuable insights on the transferability of robustness with linear probing. In Tab. 4, we find that if the AS score on the unlabeled pretraining and downstream datasets are similarly small, then it is reasonable to expect that robustness will transfer via linear probing. However, if the AS score on the downstream task is much higher, it is reasonable to expect that robustness will not transfer via linear probing.

In Tab. 4, we compare the AS score to the relative difference between clean and robust CE loss on each dataset, i.e., (CE - robust CE)/CE. Indeed, the similar AS scores of CIFAR-100, CIFAR-10, and Intel Image upon pretraining on CIFAR-100 correspond to a successful transfer of robustness, whereas the dissimilar adversarial sensitivity scores of ImageNet, Fashion-MNIST, and CIFAR-10 upon pretraining on ImageNet reflect the striking loss of robustness upon transfer. One explanation for a larger AS score is a shift in distribution between pretraining and downstream data. This connection is confirmed in an additional empirical analysis in Appx. B.5.

Table 4: For the CIFAR-100 and ImageNet pretraining datasets, we report the AS score (Eq. (2)), clean & robust cross-entropy loss, and the relative difference between said losses for the pretraining dataset, as well as the CIFAR-10, Fashion-MNIST and Intel Image downstream tasks.

| Pretraining | Downstream | AS score | clean /robust CE | relative dif. |
|---|---|---|---|---|
| CIFAR-100 | CIFAR-100 | 1.0 | 2.71 / 3.27 | 0.21 |
| | CIFAR-10 | 0.98 | 0.82 / 1.55 | 0.9 |
| | Fashion-MNIST | 2.11 | 0.45 / 1.25 | 1.8 |
| | Intel Image | 1.05 | 0.61 / 1.08 | 0.76 |
| ImageNet | ImageNet | 1.27 | 1.38 / 1.64 | 0.19 |
| | CIFAR-10 | 15.7 | 0.22 / 4.36 | 18.8 |
| | Fashion-MNIST | 53.2 | 0.36 / 9.76 | 26.1 |
| | Intel Image | 2.75 | 0.25 / 0.39 | 0.57 |

## 6.2 Beyond Linear Probing?

In the prior section, we observe a correlation between AS score and the difference between clean and robust CE, i.e., the AS score is predictive of robustness transfer via linear probing. However, linear probing is not the only transfer learning approach, what can we say about alternatives?

In finetuning, the representation function is changed, and thus also the AS score changes post-transfer. Thus, we can expect the AS score computed pre-transfer is less predictive of the downstream robustness for finetuning than for linear probing. To gain a better understanding, we study the difference between clean/robust accuracy for linear probing (LP), full finetuning (FT) as well as linear probing then finetuning (LP-FT) [35], where LP-FT was proposed to combine the benefits of the two approaches. The main results can be found in Tab. 5, and additional details about the experiment are provided in Appx. B.4.

Tab. 5 shows the performance of each transfer learning method applied to the model robustly pretrained on CIFAR-100 or ImageNet. On CIFAR-100, LP preserves the most robust accuracy, whereas LP-FT performs better on ImageNet. Note that the AS score for LP is highlighted, as this is the score we can obtain prior to transfer learning, which allows us to calibrate expectations on downstream performance.

We find no correlation between LP and FT performance, and do not observe a predictive relationship between AS score pre-transfer and robustness transfer via FT. Instead, we observe successful robustness transfer via FT and LP-FT for cases with high AS score pre-transfer. This provides us with a first indication that the adversarial sensitivity score may point to scenarios where FT preserves more robustness downstream than LP.

## 7 Discussion

**Significance:** If the representation does not change at all in response to $\epsilon$-perturbations, it may seem immediately clear that any predictor operating on said representation is robust. However, if the representation function *is* variant to $\epsilon$-perturbations, how much robustness is lost as a result is dependent on the nature of the predictor function under consideration and the distribution shift in the data from pretraining to target task. In this work, we consider a linear predictor, and formally compute the exact effect this function can have on the test-time robustness.

**Comparison of the theorems:** The theoretical statement in Thm. 1 is closely related to the one in Thm. 2, as a more robust predictor to adversarial attacks leads to a smaller difference in the loss

Table 5: Clean accuracy (Acc) and robust accuracy (RAcc) for the three transfer learning approaches using the model robustly pretrained on CIFAR-100.

| CIFAR-100 | CIFAR-10 | | | Fashion-MNIST | | | Intel Image | | |
|---|---|---|---|---|---|---|---|---|---|
| **Method** | Acc | RAcc | AS | Acc | RAcc | AS | Acc | RAcc | AS |
| FT | **94.2%** | 3.9% | 8.93 | **95.1%** | 40% | 6.68 | **90.1%** | 7.5% | 6.86 |
| LP | 70.9% | **42.2%** | **0.98** | 84% | **56.2%** | **2.11** | 78.5% | **56.8%** | **1.05** |
| LP-FT | 93.9% | 1.4% | 6.44 | 95% | 36% | 4.27 | 89.2% | 27.1% | 3.02 |
| ImageNet | CIFAR-10 | | | Fashion-MNIST | | | Intel Image | | |
| **Method** | Acc | RAcc | AS | Acc | RAcc | AS | Acc | RAcc | AS |
| FT | **98.6%** | 38.4% | 12.6 | **95.7%** | 63.5% | 9.0 | **93.9%** | 85.8% | 3.8 |
| LP | 92.4% | 15.8% | **15.7** | 87.2% | 0% | **53.2** | 91.2% | 85.3% | **2.7** |
| LP-FT | 98.5% | **44.5%** | 9.8 | 95.6% | **73.9%** | 5.1 | **93.9%** | **85.9%** | 3.9 |

between clean and adversarial examples, and vice versa. Thm. 1 provides a bound on the robustness based on the difference of the loss function, while Thm. 2 only considers the specific form of the classifier and thus information about and restrictions on the used loss function is unnecessary. However, studying the loss function makes Thm. 1 valid in a much broader set of scenarios, and further provides reasoning on why using weight norm regularization during linear probing improves robustness transfer, see Sec. 5.1.

The bound in Thm. 2, on the other hand, provides insights on how robustness of a classifier can be guaranteed. In particular, one could use our results to derive robustness guarantees post-transfer from attacks on the representation function pre-transfer.

**Clarification:** Note that we consider a model $f$ as a composition of a linear layer and a representation function, i.e., $f(x) = W g_\theta(x)$, but do not specify how the parameters of $f$ are learned. As our motivating example, the representation function $g_\theta$ is optimized on a pretraining task and subsequently $W$ is trained w.r.t. a downstream task. However, one could also approach the downstream task with the classical strategy of training the model end-to-end instead, i.e., optimizing $\tilde{W}$ and $\tilde{\theta}$ simultaneously in $f_{\tilde{W},\tilde{\theta}} = \tilde{W} g_{\tilde{\theta}}$, assuming the same architecture as before. The parameters $\theta, W$ and $\tilde{\theta}, \tilde{W}$ obtained from the two strategies may be very different. For example, the representation function $g_\theta$ may not be as useful for linearly separating the dataset into $c$ classes as $g_{\tilde{\theta}}$, which was optimized specifically for this task. This observation is described as a task mismatch between pretraining and the supervised downstream task.

Our contributed theoretical results demonstrate that we can study robustness by studying the learned functions themselves, irregardless of how said functions were optimized in the first place. With that said, this does not mean that the aforementioned task mismatch does not affect the model's robustness. Thm. 1's bound will loosen if, e.g., $g_\theta$ does not enable linear separation of the dataset into $c$ classes, and if the AS score of the representation function increases due to this task mismatch or distribution shift in the data. We provide a study on the impact of the distribution shift in the data on the AS score in Appx. B.5.

Finally, while the linear probing protocol is standard practice [10, 11], alternative methodology exists for leveraging a representation function for a specified downstream task. For one, we could consider a model $f$ as a composition of a nonlinear function $h$ and a representation function, i.e. $f(x) = h \circ g_\theta(x)$, a scenario our theoretical statements do not account for. Furthermore, alternative approaches can include updating the parameters of the representation function $g_\theta$, e.g., full finetuning. As suggested by the results in Sec. 6.2, while a high AS score may not be predictive of robustness transfer for finetuning, it may instead indicate when finetuning is required to transfer robustness. Overall, the theoretical dependencies between the robustness of the representations and transfer learning alternatives to linear probing present interesting opportunities for future analysis.

## 8 Conclusion

We prove two theorems which demonstrate that the robustness of a linear predictor on downstream tasks can be bound by the robustness of its underlying representation. We provide a result applicable to various loss functions and downstream tasks, as well as a more direct result for classification. Our

theoretical insights can be used for analysis when representations are transferred between domains, and sets the stage for carefully studying the robustness of adapting pretrained models.

**Acknowledgements**

We thank the ICLR 2022 Diversity, Equity & Inclusion Co-Chairs, Rosanne Liu and Krystal Maughan, for starting the CoSubmitting Summer (CSS) program, through which this collaboration began. We thank the ML Collective community for the ongoing support and feedback of this research. The authors thank the International Max Planck Research School for Intelligent Systems (IMPRS-IS) for supporting YS. This work was supported by the German Federal Ministry of Education and Research (BMBF): Tübingen AI Center, FKZ: 01IS18039A.

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

# A   Proofs and Auxiliary Results

We state two auxiliary results first and subsequently provide the full proofs for Thm. 1 and Thm. 2 in Appx. A.1 and Appx. A.2.

**Lemma 2.** *For a matrix $W \in \mathbb{R}^{c \times r}$ and a vector $g \in \mathbb{R}^r$, the following bounds hold,*

1. $\|Wg\|_2 \leq \|W\|_2 \|g\|_2$,

2. $\|Wg\|_1 \leq \sum_{i=1}^{c} \|W_i\|_2 \|g\|_2$,

3. $\|Wg\|_\infty \leq \max_i \|W_i\|_2 \|g\|_2$.

*Proof.*    1. $\|Wg\|_2 = \sqrt{\sum_{i=1}^{c} (W_i g)^2} \leq \sqrt{\sum_{i=1}^{c} (\|W_i\|_2 \|g\|_2)^2} = \|W\|_2 \|g\|_2$, where the second step follows by Cauchy-Schwartz inequality.

2. $\|Wg\|_1 = \sum_{i=1}^{c} |W_i g| \leq \sum_{i=1}^{c} \|W_i\|_2 \|g\|_2 = \sum_{i=1}^{c} \|W_i\|_2 \|g\|_2$, where the second step follows again by Cauchy-Schwartz inequality.

3. $\|Wg\|_\infty = \max_i |W_i g| \leq \max_i \|W_i\|_2 \|g\|_2$, where the second step follows by Cauchy-Schwartz inequality.

$\square$

The following Lemma proves a Lipschitz condition for the cross-entropy-softmax loss.

**Lemma 3.** *We define the cross-entropy loss with softmax function for a vector $f \in \mathbb{R}^c$ and a class index $y \in \{1, \ldots, c\}$, i.e.,*

$$\ell(f, y) := -\log(p_y(f)) = -\log\left(\frac{\exp\{f_y\}}{\sum_{i=1}^{c} \exp\{f_i\}}\right).$$

*Then, the following Lipschitz condition holds for $\ell(f, y)$,*

$$|\ell(a, y) - \ell(b, y)| \leq 2 \|a - b\|_\infty,$$

*for any vectors $a, b \in \mathbb{R}^c$ and an arbitrary class index $y \in \{1, \ldots, c\}$.*

*Proof.*

$$
\begin{aligned}
|\ell(a, y) - \ell(b, y)| &= \left| -\log\left(\frac{\exp\{a_y\}}{\sum_{j=1}^{c} \exp\{a_j\}}\right) + \log\left(\frac{\exp\{b_y\}}{\sum_{j=1}^{c} \exp\{b_j\}}\right) \right| \\
&= \left| \log\left(\exp\{b_y - a_y\}\right) - \log\left(\frac{\sum_{j=1}^{c} \exp\{b_j\}}{\sum_{j=1}^{c} \exp\{a_j\}}\right) \right| \\
&= \left| (b_y - a_y) - \log\left(\sum_{j=1}^{c} \frac{\exp\{a_j\}}{\sum_{l=1}^{c} \exp\{a_l\}} \exp\{b_j - a_j\}\right) \right| \\
&\leq \left| (b_y - a_y) - \sum_{j=1}^{c} p_j(a)(b_j - a_j) \right|,
\end{aligned}
$$

where $p_j(a) := \frac{\exp\{a_j\}}{\sum_{l=1}^{c} \exp\{a_l\}}$ denotes the soft-max output of the vector $a$. The inequality in the last line follows by Jensen's inequality, since $-\log$ is convex and the soft-max probabilities $p_j$ sum up to one.

$$\left| (b_y - a_y) - \sum_{j=1}^{c} p_j(a)(b_j - a_j) \right| \leq |b_y - a_y| + \sum_{j=1}^{c} p_j(a)|b_j - a_j| \leq 2 \|b - a\|_\infty,$$

since both summands in the middle can be bounded by the supremum norm.    $\square$

## A.1 Proof of Theorem 1

*Proof of Theorem 1.* We use Hoeffding's inequality on the independent random variables $D_1, \ldots, D_n$, which are defined as

$$D_i := \max_{\|\delta\| < \epsilon} \ell(f(x_i + \delta), y_i) - \ell(f(x_i), y_i),$$

based on independently drawn data points with probability distribution $\mathcal{P}(\mathcal{X})$, where the observations are $x_i \in \mathcal{X}$ with corresponding labels $y_i$. We can use the upper bound on $\ell$ to obtain that $0 \le D_i \le C_2$ and conclude that

$$\mathbb{P}\left(\left|\sum_{i=1}^n D_i - n\mathbb{E}[D]\right| \ge t\right) \le 2 \cdot \exp\left(\frac{-2t^2}{nC_2^2}\right)$$

$$\Rightarrow \quad \mathbb{P}\left(\left|\frac{1}{n}\sum_{i=1}^n D_i - \mathbb{E}[D]\right| \le C_2\sqrt{\frac{\log(\rho/2)}{-2n}}\right) \ge 1 - \rho.$$

Thus, with probability of at least $1 - \rho$ it holds that

$$\mathbb{E}[D] = \left|\mathcal{L}_{\text{adv}}(f) - \mathcal{L}(f)\right| = \left|\mathbb{E}_{(x,y)}\left[\max_{\|\delta\|<\epsilon} \ell(f(x+\delta), y) - \ell(f(x), y)\right]\right|$$

$$\le \left|\frac{1}{n}\sum_{i=1}^n \max_{\|\delta\|<\epsilon} \ell(f(x_i+\delta), y_i) - \ell(f(x_i), y_i)\right| + C_2\sqrt{\frac{\log(\rho/2)}{-2n}}.$$

We can further bound the first term on the right-hand side, since the loss function $\ell(r, y)$ is $C_1$-Lipschitz in $\|\cdot\|_\alpha$ for $r \in f(\mathcal{X})$ and $f(x) = W g_\theta(x)$.

$$\left|\frac{1}{n}\sum_{i=1}^n \max_{\|\delta\|<\epsilon} \ell(W g_\theta(x_i+\delta), y_i) - \ell(W g_\theta(x_i), y_i)\right|$$

$$= \left|\frac{1}{n}\sum_{i=1}^n \left|\ell(W g_\theta(x_i+\delta_i), y_i) - \ell(W g_\theta(x_i), y_i)\right|\right|$$

$$\le C_1 \frac{1}{n}\sum_{i=1}^n \|W g_\theta(x_i+\delta_i) - W g_\theta(x_i)\|_\alpha,$$

where $\delta_1, \ldots, \delta_n \in \mathcal{X}$ with $\|\delta_i\| < \epsilon$ are chosen appropriately. In a next step, we apply Lemma 2 and the definition of $L_\alpha$ in the theorem statement,

$$C_1 \frac{1}{n}\sum_{i=1}^n \|W g_\theta(x_i+\delta_i) - W g_\theta(x_i)\|_\alpha \le L_\alpha(W) C_1 \frac{1}{n}\sum_{i=1}^n \|g_\theta(x_i+\delta_i) - g_\theta(x_i)\|_2.$$

The $\delta_i$'s might not be the same that maximize the distance between the representations but their effect can be upper bounded by the maximum. Combining the derived bounds and using that $g_\theta(x_i+\delta_i) \le \max_{\|\delta\|<\epsilon} g_\theta(x_i+\delta)$, we obtain our result.

$$\mathcal{L}_{\text{adv}}(f) - \mathcal{L}(f) = \left|\mathcal{L}_{\text{adv}}(f) - \mathcal{L}(f)\right|$$

$$\le L_\alpha(W) C_1 \frac{1}{n}\sum_{i=1}^n \max_{\|\delta\|<\epsilon} \|g_\theta(x_i+\delta) - g_\theta(x_i)\|_2 + C_2\sqrt{\frac{\log(\rho/2)}{-2n}}.$$

$\square$

## A.2 Proof of Theorem 2

*Proof of Theorem 2.* We want to proof that the classifier $c(x)$ is robust in $x \in \mathcal{X}$, i.e., it holds that $c(x+\delta) = c(x)$ for all $\delta \in \mathbb{R}^d$ with $\|\delta\| < \epsilon$. The classifier predicts class $y$, i.e., $c(x) = y$, if the y-th value of the model $f(x)$ is larger than all other entries, i.e., $f(x)_y > f(x)_j$ for all $j \ne y$. We

denote the y-th row of the linear layer's matrix $W$ by $W_y$, thus $f(x)_y = W_y g_\theta(x)$.
For an input $x$ with $c(x) = y$, we want to prove that $c(x + \delta) = y$ for any $\delta$ with $\|\delta\| < \epsilon$., i.e.,

$$f(x + \delta)_y > f(x + \delta)_j, \ \text{ if } \ f(x)_y > f(x)_j.$$

It holds that

$$f(x + \delta)_y = f(x + \delta)_j + (W_j(g_\theta(x) - g_\theta(x + \delta))) + (W_y g_\theta(x) - W_j g_\theta(x)) + (W_y(g_\theta(x + \delta) - g_\theta(x))).$$

Thus, for $f(x + \delta)_y > f(x + \delta)_j$ to be true, it is necessary that

$$(W_j(g_\theta(x) - g_\theta(x + \delta))) + (W_y g_\theta(x) - W_j g_\theta(x)) + (W_y(g_\theta(x + \delta) - g_\theta(x))) \geq 0$$
$$\iff \quad W_y g_\theta(x) - W_j g_\theta(x) \geq (W_y - W_j)(g_\theta(x) - g_\theta(x + \delta))). \tag{3}$$

By Cauchy-Schwartz inequality follows that

$$(W_y - W_j)(g_\theta(x) - g_\theta(x + \delta))) \leq |(W_y - W_j)(g_\theta(x) - g_\theta(x + \delta))| \leq \|W_y - W_j\|_2 \cdot \|g_\theta(x + \delta) - g_\theta(x)\|_2.$$

Since further $W_y g_\theta(x) - W_j g_\theta(x) = f(x)_y - f(x)_j$, the condition of equation (3) holds for any $j \neq y$ if

$$f(x)_y - f(x)_j = |f(x)_y - f(x)_j| \geq \|W_y - W_j\|_2 \cdot \|g_\theta(x + \delta) - g_\theta(x)\|_2.$$

For $c(x + \delta) = c(x)$ to be true, the above condition needs to be true for all $j \neq y$ and this holds if

$$\min_{j \neq y} \frac{|f(x)_y - f(x)_j|}{\|W_y - W_j\|_2} \ \geq \ \max_{\|\delta\| < \epsilon} \|g_\theta(x + \delta) - g_\theta(x)\|_2.$$

$\square$

# B  Additional Experimental Results

We provide additional information on the datasets used and the employed hyperparameters in Appx. B.1. A further application of Thm. 1 is presented in Appx. B.2, where robustness transfer is analysed for a regression task. In Appx. B.3, we extend the analysis from Sec. 6 by evaluating the robustness against stronger attacks. In Appx. B.4, an empirical study compares the robustness transfer depending on the chosen transfer learning method. Finally, in Appx. B.5, we investigate the effect of a shift in distribution between pretraining and downstream task data on the transferred robustness and the AS score.

## B.1  Experimental Setup for Section 5

For our experiments, we consider two models robustly pretrained on CIFAR-100 [34] and ImageNet [17]. CIFAR-100 consists of 32x32 RGB images with 100 different classes. ImageNet-1K consists of 256x256 RGB images of 1,000 different classes.
We investigate the performance of robustness transfer on CIFAR-10 [34], Fashion-MNIST [57] and Intel Image Classification[3]. The CIFAR-10 dataset consists of 32x32 RGB images in 10 classes. Intel Image Classification consists of 150x150 RGB images of 6 natural scenes (buildings, forest, glacier, mountain, sea, and street). Fashion-MNIST is a dataset of Zalando's article images, where each example is a 28x28 grayscale image, associated with a label from 10 classes.

We use two models from RobustBench [16], namely robust models trained on CIFAR-100 and ImageNet. For CIFAR-100, we use a WideResNet-34-10 ($\approx$46M parameters) architecture [61] which is adversarially trained using a combination of simple and complex augmentations with separate batch normalization layers [1]. For ImageNet, we use a WideResNet-50-2 ($\approx$66M parameters) architecture [61] which is is also adversarially trained [47, 40].

For the model pretrained on CIFAR-100, run 20 epochs of linear probing (LP) using a batch size of 128, and resize all input images to 32x32. For the model pretrained on ImageNet, we run 10 epochs of LP using a batch size of 32, and resize all input images to 256x256. We reduce the batch size to compensate for the increase in input resolution. For LP, we set the learning rate using a cosine

---

[3] https://www.kaggle.com/datasets/puneet6060/intel-image-classification

| Layer | Layer Type | Activation and Normalization |
|-------|-----------|------------------------------|
| 1 | Convolutional layer (c=6, k=5) | ReLu, Average Pooling, and Batch Norm |
| 2 | Convolutional layer (c=16, k=5) | ReLu, Average Pooling, and Batch Norm |
| 3 | Linear Layer (out=120) | ReLu and Batch Norm |
| 4 | Linear Layer (out=84) | ReLu and Batch Norm |
| 5 | Linear Layer (out=84) | ReLu and Batch Norm |
| "head" | Linear Layer (out=1) | - |

Table 6: The table presents the model architecture used in experiments on MSE loss and the dSprites dataset. The convolutional layers are provided with number of output channels and kernel size, i.e., (c=channels, k=kernel_size), where as the linear layers are listed together with the size of their output vector.

annealing schedule [39], with an initial learning rate of 0.01, and use stochastic gradient descent with a momentum of 0.9 on the cross-entropy loss.

To compute LHS and RHS, we must attack the cross-entropy loss on $f$ as well as the MSE on $g_\theta$. In both cases, we employ $L_\infty$-PGD attacks implemented in the foolbox package [45]. As hyperparameters for the $L_\infty$-PGD attack, we choose 20 steps and a relative step size of 0.7, since this setting yields highest adversarial sensitivity (AS) scores over all the tasks. The attack strength $\epsilon$ is set to $\epsilon = 8/255$ for attacking the CIFAR-100 pretrained model and $\epsilon = 1/255$ for the ImageNet model. Thus, we use the same $\epsilon$ on CIFAR-100 as used during the RobustBench pre-training and reduce the $\epsilon$ on ImageNet due to the constant pixel blocks introduced after resizing CIFAR-10 and Fashion-MNIST images from 32x32 and 28x28 to 256x256, respectively.

The experiments were run on a single GeForce RTX 3080 GPU and for each downstream task transfer learning or theory evaluation took between 30 minutes to 3 hours.

## B.2 Regression - MSE

Here, we empirically evaluate Lemma 1 on regression tasks using the mean squared error (MSE) as the scoring criterion. As per Sec. 3.1, we can apply Lemma 1 with Lipschitz constant $C = 1$ and mapping function $L_2(W)$.

We consider the dSprites [41] dataset, which consists of 64x64 grayscale images of 2D shapes procedurally generated from 6 factors of variation (FoVs) which fully specify each sprite image: color, shape, scale, orientation, x-position (posX) and y-position (posY). The dataset consists of 737,280 images, of which we use an 80-20 train-test split. Here, we pre-train for the regression task of decoding an FoV from an image, i.e., we optimise the network $f$ to minimize MSE with the exact orientation of the sprite. For LP, we consider 4 downstream tasks, each corresponds to decoding a unique FoV, while one matches the task used for pre-training.

The model architecture is summarized in Tab. 6. We consider 3 different representation functions, network before training (with default kaiming uniform initialization [24]), after training, and after adversarial training. For the robust model, we utilize $L_\infty$-PGD attacks implemented in the foolbox package [45] with 20 steps and a relative step size of 1.0. The attack strength $\epsilon$ is set to $\epsilon = 8/255$. The model was trained for 10 epochs with a batch size of 512, learning rate of $1e-3$ with a cosine annealing schedule [39], and stochastic gradient descent with a momentum of 0.9.

In Tab. 7, we present the left-hand side (LHS), the rescaled difference between clean and robust cross-entropy (CE) loss, and the right-hand side (RHS), the representation function's AS score, of Lemma 1. To obtain the scores for LHS and RHS, we separately attack $f$ as well as the MSE on $g_\theta$, where in both cases the objective uses the MSE. For these attacks, we again utilize $L_\infty$-PGD, but also $L_2$-PGD attacks, both based on the foolbox [45] implementation, with 50 steps and a relative step size of 1.0. The attack strength $\epsilon$ is set to $\epsilon = 16/255$ for both the attack types. As can be seen, LHS is indeed always upper bounded by RHS. Note that, for each pretraining and attack type combination, the RHS is the same over all latent factors, i.e. in each row, because the same images and the same representation function are used.

| Pretraining | Attack Type | | Scale | Orientation | posX | posY |
|---|---|---|---|---|---|---|
| Adversarial | $L_\infty$-PGD | LHS | 0.10 | 0.09 | 0.02 | 0.02 |
| | | RHS | 2.29 | 2.29 | 2.29 | 2.29 |
| Standard | $L_\infty$-PGD | LHS | 3.32 | 1.20 | 2.02 | 2.95 |
| | | RHS | 23.80 | 23.80 | 23.80 | 23.80 |
| Random | $L_\infty$-PGD | LHS | 2.33 | 2.29 | 2.43 | 1.84 |
| | | RHS | 29.25 | 29.25 | 29.25 | 29.25 |
| Adversarial | $L_2$-PGD | LHS | 0.004 | 0.003 | 0.001 | 0.001 |
| | | RHS | 0.15 | 0.15 | 0.15 | 0.15 |
| Standard | $L_2$-PGD | LHS | 0.33 | 0.22 | 0.17 | 0.19 |
| | | RHS | 3.15 | 3.15 | 3.15 | 3.15 |
| Random | $L_2$-PGD | LHS | 0.19 | 0.21 | 0.14 | 0.12 |
| | | RHS | 1.69 | 1.69 | 1.69 | 1.69 |

Table 7: Left-hand side (LHS) and right-hand side (RHS) of Lemma 1 using a model pretrained on the *orientation* latent factor of dSprites.

### B.3 Classification Criterion - Attack Strength

Here, we study the hypothesis stated in Sec. 5.2 that datapoints fulfilling the bound in Thm. 2 are still robust against an increased attack strength. We repeat the experiments from Sec. 5.2 for the model pretrained on CIFAR-100, where the attack strength is $\epsilon = 8/255$, since, for this case, a subset of each dataset fulfilled the equation. Additionally, we compute the robust accuracy against stronger attacks employing double and three times the attack strength, i.e., using $2\epsilon = 16/255$ and $3\epsilon = 24/255$.

In Tab. 8, we observe that indeed most of the datapoints fulfilling the bound in Eq. (1) are still robust against the doubled attack strength, which is suggested by the robust accuracy for $2\epsilon$ being larger than the proportion that fulfilled the bound. Most of these datapoints, approximately $50\%$, are even robust against $3\epsilon$. Further analysis is needed to better understand the observed robust accuracy scores and to derive a precise connection to the bound from Thm. 2.

| | | | CIFAR-100 | |
|---|---|---|---|---|
| | attack strength | CIFAR-10 | Fashion-MNIST | Intel Image |
| clean accuracy | 0 | 70.9% | 84% | 78.5% |
| robust accuracy | $\epsilon$ | 42.2% | 56.2% | 56.8% |
| prop. fulfilled | $\epsilon$ | 11.3% | 4.6% | 27.1% |
| robust accuracy | $2\epsilon$ | 18.4% | 28.5% | 31.4% |
| robust accuracy | $3\epsilon$ | 6.6% | 13.1% | 13.4% |

Table 8: For the CIFAR-100 pretraining dataset, we report clean accuracy, the proportion of images fulfilling the theoretical bound, and robust accuracy for several attack strengths for each of the CIFAR-10, Fashion-MNIST and Intel Image downstream tasks.

### B.4 Comparison Study between LP, FT and LP-FT

Here, we compare robustness transfer on classification tasks for three transfer learning techniques, linear probing, full finetuning, and linear probing then finetuning [35]. As explained before, linear probing (LP) refers to freezing all pretrained weights and only optimizing the linear classification head. In full finetuning (FT) all pretrained parameters are updated during transfer learning. Linear probing then finetuning (LP-FT) combines LP and FT sequentially, as after LP is done, FT is applied to all parameters.

For the model pretrained on CIFAR-100, we choose a batch size of 128, and run 20 epochs for FT, LP and 10 epochs of FT after LP for LP-FT. For the model pretrained on ImageNet, we choose a batch size of 32 to compensate for the increase in input resolution to 256x256. We run 10 epochs for FT and LP and 5 epochs for LP-FT. In both settings, we use half as many epochs for finetuning

after linear probing in LP-FT, as in [35]. For all methods we apply a cosine annealing learning rate schedule [39], with an initial learning rate of 0.01 for LP on CIFAR-100, and 0.001 for all other scenarios. We use stochastic gradient descent with a momentum of 0.9 on the cross-entropy loss for all experiments.

We evaluate the adversarial robustness on the test set after training by employing the $L_\infty$-PGD attack provided by the foolbox package [45], with 20 steps and a relative step size of 0.7. As the attack strength, we choose $\epsilon = 8/255$ for the model pretrained on CIFAR-100. Due to the increase in input resolution for the ImageNet model, we apply $\epsilon = 1/255$.

Tab. 5 in Sec. 6.2 shows the performance of each transfer learning method applied to the model robustly pretrained on CIFAR-100. For all datasets, LP preserves the most robust accuracy, while performing worst on natural examples. FT and LP-FT show similar natural and robust accuracy. Only for the Intel Image dataset do we observe a significant difference (10%) in the robustness of LP-FT and FT. In contrast, when pretraining on ImageNet, LP has the worst clean and robust accuracy. Instead, LP-FT has the highest robust accuracy for all datasets, while FT achieves the highest clean accuracy. Although FT has the highest clean accuracy for all datasets, the difference between FT and LP-FT is marginal.

Additionally, Tab. 9 provides confirmation that our theory indeed holds for the other transfer learning approaches as well. The difference is that for FT and LP-FT the AS score changes during transfer learning and thus the score computed prior transfer learning is no longer predictive of the transferred robustness. Still, it is an interesting observation that indeed the AS score shrinks during LP-FT when it improves robustness transfer compared to LP.

| CIFAR-100 | CIFAR-10 | | | Fashion-MNIST | | | Intel Image | | |
|---|---|---|---|---|---|---|---|---|---|
| **Method** | $\Delta$CE | $L_\infty(W)$ | AS | $\Delta$CE | $L_\infty(W)$ | AS | $\Delta$CE | $L_\infty(W)$ | AS |
| FT | 8.7 | 1.31 | 8.93 | 4.49 | 1.26 | 6.68 | 5.62 | 1.1 | 6.86 |
| LP | 0.73 | 4.4 | 0.98 | 0.8 | 4.59 | 2.11 | 0.47 | 3.05 | 1.05 |
| LP-FT | 14.1 | 4.42 | 6.44 | 5.98 | 4.6 | 4.27 | 3.06 | 3.07 | 3.02 |
| ImageNet | CIFAR-10 | | | Fashion-MNIST | | | Intel Image | | |
| **Method** | $\Delta$CE | $L_\infty(W)$ | AS | $\Delta$CE | $L_\infty(W)$ | AS | $\Delta$CE | $L_\infty(W)$ | AS |
| FT | 4.39 | 1.1 | 12.6 | 2.28 | 1.1 | 9.0 | 0.32 | 0.94 | 3.8 |
| LP | 4.14 | 1.86 | 15.7 | 9.4 | 2.02 | 53.2 | 0.14 | 1.39 | 2.7 |
| LP-FT | 3.18 | 1.9 | 9.8 | 0.92 | 2.06 | 5.1 | 0.37 | 1.0 | 3.9 |

Table 9: Absolute difference between clean and robust CE loss ($\Delta$CE ), $L_\infty(W)$ from Thm. 1 and the adversarial sensitivity score (AS) for the three transfer learning approaches using the model robustly pretrained on CIFAR-100 or ImageNet.

## B.5 Distribution Shift Experiments

In this section, we analyse the impact of distribution shift between pretraining and downstream data on the adversarial sensitivity score and thereby on robustness transfer via linear probing. First, we consider the distribution shift between CIFAR-10 and ImageNet and, in particular, its dependence on how the smaller CIFAR-10 images are resized before applying the representation function. Second, we employ ideas from prior work [26][4] to create increasingly corrupted versions of CIFAR-100 to simulate varying shifts in distribution.

In our experiments in Sec. 5, we resize the 32x32 CIFAR-10 input images to 256x256 for the ImageNet model, and observe a strong distribution shift by a severely increased AS score. Here, we evaluate if increasing the image size before applying the ResNet model causes this shift, or if it is inherent in the data. In Tab. 10, we repeat our linear probing experiment for the CIFAR-10 to ImageNet case on differently rescaled input images. The results indicate that reducing the increase in the input image size improves robustness transfer, at the cost of a drop in clean performance. However, even for a smaller input image size, the AS score remains increased indicating a tangible distribution shift remains.

---

[4]https://github.com/hendrycks/robustness/blob/master/ImageNet-C/create_c

| input image | clean /robust CE | clean /robust acc | prop. fulfilled | LHS | RHS |
|---|---|---|---|---|---|
| 256x256 | 0.22 / 4.36 | 92% / 16% | 0.1% | 1.14 | 15.7 |
| 128x128 | 0.41 / 1.31 | 86% / 57% | 0.2% | 0.28 | 12.7 |
| 64x64 | 1.04 / 1.49 | 64% / 48% | 0.1% | 0.14 | 8.2 |

Table 10: The table contains clean and robust CE loss and accuracy, the proportion of images fulfilling the theoretical bound in Eq. (1), LHS and RHS of Lemma 1, when performing linear probing from an ImageNet pre-trained model based on differently resized CIFAR-10 images.

In the following, we consider an increasingly corrupted version of CIFAR-100 to simulate distribution shift following prior work [26]. In particular, we use the Gaussian noise (Tab. 12) and Zoom blur (Tab. 13) corruption types with increasing severity to simulate an increasing distribution shift. The experimental results from both distortion methods confirm that an increasing shift between the data distributions leads to an increase in the AS score and poorer robustness transfer via linear probing. The different corruption strength are used as suggested in [26] and are stated in Tab. 11.

| **Approach** | C1 | C2 | C5 |
|---|---|---|---|
| Gaussian Noise | 0.04, | 0.06 | 0.10 |
| Zoom Blur | [1, 1.06] | [1, 1.11] | [1, 1.26] |

Table 11: Definition of the corruption strength considered in our experiments. The Gaussian noise corruption severity is defined by the scale, i.e. the variance of the Gaussian noise distribution. The Zoom blur is defined via the interval of zoom factors. For each 0.01 increment in the interval, a zoomed-in image is created, and all zoomed-in images are summed to create a blurred image.

| **Metric** | CIFAR-100 | CIFAR-100-C1 | CIFAR-100-C2 | CIFAR-100-C5 |
|---|---|---|---|---|
| Acc. | 67.3% | 66.5% | 65.3% | 63.6% |
| Robust Acc. | 46.8% | 44.5% | 43.3% | 40.9% |
| CE | 1.28 | 1.32 | 1.37 | 1.44 |
| robust CE | 2.1 | 2.21 | 2.3 | 2.44 |
| difference CE | 0.82 | 0.89 | 0.93 | 1.0 |
| $L_\infty(W)$ | 3.31 | 3.34 | 3.36 | 3.47 |
| LHS (Lem.1) | 0.124 | 0.133 | 0.138 | 0.144 |
| AS score | 0.835 | 0.892 | 0.915 | 0.949 |

Table 12: Various metrics for transfer learning from a model robustly pretrained on CIFAR-100 to an increasingly corrupted CIFAR-100 dataset using Gaussian noise. C1 indicates severity level 1 of the corruption is applied.

| Metric | CIFAR-100 | CIFAR-100-C1 | CIFAR-100-C2 | CIFAR-100-C5 |
|---|---|---|---|---|
| Acc. | 67.3% | 65.4% | 65.7% | 63.9% |
| Robust Acc. | 46.8% | 41.2% | 40.3% | 35.5% |
| CE | 1.28 | 1.36 | 1.36 | 1.43 |
| robust CE | 2.1 | 2.39 | 2.42 | 2.73 |
| difference CE | 0.82 | 1.03 | 1.06 | 1.3 |
| $L_\infty(W)$ | 3.31 | 3.36 | 3.35 | 3.42 |
| LHS (Lem.1) | 0.124 | 0.153 | 0.158 | 0.19 |
| AS score | 0.835 | 1.001 | 1.031 | 1.195 |

Table 13: Various metrics for transfer learning from a model robustly pretrained on CIFAR-100 to an increasingly corrupted CIFAR-100 dataset using Zoom blur. C1 indicates severity level 1 of the corruption is applied.

