# OpenReview forum: "On Transfer of Adversarial Robustness from Pretraining to Downstream Tasks"
_NeurIPS.cc/2023/Conference — NeurIPS 2023 poster_

### Official Review · Reviewer_MCBm · 2023-06-29

**Soundness:** 2 fair
**Presentation:** 3 good
**Contribution:** 1 poor
**Rating:** 3
**Confidence:** 4

**Summary:**

This paper studies the question of how the robustness of pretrained representations transfers to downstream tasks. Under the linear probing setting (i.e., the downstream classifier $f$ is linearly related to the pretrained representations $g_{\theta}$), they prove that the gap between the standard and robust risk of $f$ can be bounded by the summand of the "perturbation on representations" plus a constant (w.r.t. $g_{\theta}$ and $x$). By rearranging terms, they obtain the *robustness score* of a dataset, which can be used to estimate how well the robustness of pretraining would transfer to downstream tasks. This paper also provides an upper bound for the robustness score. Experiments on several datasets are conducted to validate their theoretical results.

**Strengths:**

1. This paper provides theoretical results that upper bounds the gap between robust and standard loss under the setting that they concern. Besider, the experimental results perfectly fit their theory.
2. They propose the concept of robustness score, which is easy to calculate and is useful in estimating the robustness of the downstream classifiers.


**Weaknesses:**

+ The main theoretical result (i.e., Theorems 1) is essentially a robust risk bound for linear classifiers, which is, technically speaking, quite a simple result. Previous works have proved similar robust risk bounds for 2-layer ReLU networks and even more generally, for VC-class functions. I understand that Theorem 1 indeed provides a robust risk bound for downstream tasks under the linear probing setting, and the analyse for linear classifiers cannot be extended to other function classes due to linear probing. However, I still think the present paper's theoretical contribution is not enough.
+ **The current version of this paper looks like a work in progress.** As is stated above, the theoretical contribution of this paper is not enough. Maybe the author should complete the future work that is mentioned in Section 7.
+ The writing of this paper might need further polishing. For example, Figure 1 does not illustrates the intuition behind their theory (something related to your proofs is expected). The related work is also hard to understand, making the readers hard to evaluate the contribution of this paper.


**Questions:**

**Section 2 (Related Work) is hard to understand.** Could the author please explain, in more detail, the relationship between the related works and yours? My major questions include:

1. In Line 55-59, the author claims that ref. [54, 43, 15, 36, 35, 45], instead of ref. [25], provides practical guarantees on the detection of adversarial examples. By saying "instead", the author seems to be expressing the idea that ref. [25] theoretically guarantees the detection of adversarial examples. However, ref. [25] does not seem to provide such results.
2. What is the relationship between adversarial training, self-supervised contrastive learning, and the present work?
3. I don't think the present work is "in contrast with" ref. [59] and [8], as is claimed in Line 69-71. In my opinion, ref. [59], [8] and the present work are studying difference topics.
4. As is mentioned in Line 75-76 and 47-48, linear probing seems to be closely related to the main contributions of this paper. In fact, the theoretical results in Sections 3 and 4 are all based on the linear probing (i.e., $f(x) = W g_{\theta}(x)$). However, I have found no related work on linear probing in this paper. It would make this paper more comprehensible if the author briefly introduce the basic setting and related work about linear probing.


**Limitations:**

No.

---

> ### Author Rebuttal · Authors · 2023-08-09
>
> We want to thank you for your feedback and in the following we address the weaknesses and questions raised in your review.
>
> **Weaknesses:**
> 1. We are aware of results on robust risk bounds for linear classifiers or VC-class functions, which, roughly speaking, provide insights into the learnability of a robust classifier based on the function class. However, we are not aware of a theoretical result that provides similar insights into robustness transfer or could be directly applied to the scenario that we consider. Our result proves for any fixed function/network how robustness can be transferred through a linear layer and a loss function, which makes it directly applicable to many transfer learning use cases. In particular, our results are valid for several loss functions, i.e., several tasks can be considered, not just classification, and also different adversarial attacks are allowed, e.g. L_1, L_2, or L_inf attacks.
> In all, we believe that the result is novel and provides a relevant contribution to studying robustness in transfer learning. In particular, if the result is seen as “simple”, that can be advantageous for facilitating the use of our bound in applications. With that said, if we are missing some relevant theoretical contributions to this topic, it would be great if you could share the corresponding references.
> 2. Following your suggestion, we reviewed again the future work that we have suggested in section 7, and we agree that some of the mentioned topics should be included in the paper, and we will do so in the revision. In particular, we will incorporate the results and discussion mentioned below.
>     - **a)** Regarding “deriving conditions for optimal weights of the linear classifier based on the theorem’s bound”, according to our theoretical bound, regularizing the weight matrix of the linear classifier is optimal for robustness transfer, as the smaller the norm of the weight matrix, the more robustness transfer will be improved. For empirical affirmation, we provide new experimental results in Table 1 in the added pdf. In this new experiment, we repeat the experiments from Table 1 from our article, where CIFAR-100 is the pre-training set, but with regularization added during optimization ($\lambda=0.01$). The results indicate that reducing the weight norm does indeed reduce the difference between clean and robust loss, i.e., better robustness transfer.
>     - **b)** Regarding “the impact of the distribution shift in the data on the robustness score”, we will add a discussion on our experiments that already include scenarios with a strong distribution shift. For example, there is a severe distribution shift when doing transfer learning from ImageNet to F-MNIST. In this case, we can observe that the robustness score increases compared to the case with no distribution shift, i.e., ImageNet -> ImageNet, see Table 3 for the robustness scores. Thus, our empirical results imply that a larger distribution shift between pre-training and downstream tasks does lead to a larger robustness score. Additionally, we will incorporate a more granular analysis of the effect of distribution shift by performing transfer learning from ImageNet to ImageNet corrupted with variable corruption strength to obtain a fine-grained measure of the effect distribution shift has on robustness score. With that said, a theoretical analysis of the connection between robustness transfer and distribution shift remains open.
>      - **c)** Regarding “theoretical dependencies between the robustness of the representations and transfer learning alternatives to linear probing”, studying the relation between distribution shift and robustness transfer theoretically may yield insights, given our aforementioned experiments in Section 5 do suggest a connection between distribution shift and the most robustness-preserving transfer learning method.
> 3. While we agree that Figure 1 isn’t specific to our proofs, the illustration aims to show how the difference between robust and non-robust representation function, i.e., their sensitivity wrt. adversarial perturbation can affect the robustness of a linear classifier trained upon this feature representation space. The graphic intends to provide a general intuition regarding the overall subject of study than a precise insight into our theoretical contribution. While Figure 2 provides intuition for understanding one of our theoretical results, we will consider a more proof-related figure for the revision. \
> Thank you for your detailed feedback on our related work section. We answered your questions in the following and will incorporate clearer descriptions and more details on the relation to our work when revising this section.
>
> **Questions:**
> 1. We correct the writing to make this comparison more clear. Ref. [25], in Theorem 2.1, provides formal robustness guarantees for a classifier f, by leveraging a connection to Lipschitzness, but it indeed is not directly comparable to the approaches in Ref. [54, 43, 15, 36, 35, 45].
> 2. These sentences were introducing the landscape of adversarial robustness and were not intended to be directly linked to our contribution. However, we understand why a relationship was expected, and will consider this when revising this section.
> 3. Thanks for bringing this to our attention, we will correct the language.
> 4. Linear probing is studied empirically in [34] in the context of out-of-domain performance, and in [49] in the context of adversarial robustness. Our work provides theoretical justification for some of their empirical observations on linear probing. As you pointed out correctly, we should have stated this connection to our work more precisely and will provide a self-contained section to introduce linear probing to clarify where our contributions sit in that area of literature.

---

> > ### Comment · Reviewer_MCBm · 2023-08-16
> > **Thanks for your reply**
> >
> > Thanks the authors for their response. The rebuttal has addressed most of our concerns. However, I am not convinced by the response to Weakness 1.
> >
> > In the response to Weakness 1, the authors claim that "they are not aware of a theoretical result that could be directly applied to the scenario that we consider". We find some related works ([1], [2]) that provide such results. I think that Theorem 1 can be viewed as a simple corollary of the results in [1]. Consider the first equation in Theorem 1 (here, I suggest the authors to number the equations in this paper). The second term seems to be a complexity term (e.g., Rademacher complexity). [1] provides a more detailed discussion on the Rademacher complexity in the robust linear classifier setting. I wonder what is the difference between your results and theirs. More specifically, could the authors please either
> > verify that the results in this paper are in superior to those in [1] and [2], or
> > prove that the results in [1] cannot be directly apply to your setting and explain what is the gap between your setting and those of [1] and [2].
> > I would be very happy to raise my score if the authors could provide convincing response.
> >
> > Reference
> >
> > [1] Pranjal Awasthi, Natalie S. Frank, Mehryar Mohri, Adversarial Learning Guarantees for Linear Hypotheses and Neural Networks, ICML, 2020.
> > [2] Dong Yin, Kannan Ramchandran, and Peter Bartlett, Rademacher Complexity for Adversarially Robust Generalization, CoRR abs/1810.11914, 2018

---

> > > ### Author Response · Authors · 2023-08-18
> > >
> > > Thank you for your response and continuing this very important discussion to clarify our contributions. We took the time to read the two papers you referenced and, in the following, we describe the difference we see between their results and ours.
> > >
> > > Firstly, note that the second term in the first equation of our Theorem 1 is not a complexity term but a bound on the difference between clean and adversarial empirical loss. Our result does not contain such a complexity term, as the result holds for a fixed linear function w instead of uniformly over all linear functions.
> > >
> > > To make the difference more clear, we follow your suggestion and discuss a possible corollary from the results in [1] for linear probing on a pretrained representation function $g_{\theta}$. We expect that this can be done by considering the input sample $(g_{\theta}(x_1),y_1),\dots, (g_{\theta}(x_m),y_m)$ instead of $(x_1,y_1),\dots, (x_m,y_m)$. Note that we did not have time to write down a formal proof but we consider a statement similar to Theorem 1 in [1] to be reasonable. Let's consider the following holds with probability $1-\delta$ uniformly over all linear functions $w\in F$:
> > >
> > > $$  \mathbb{E} \left[ \underset{\Vert \delta\Vert <\epsilon}{\mathrm{max}}\, \ell(w(g_{\theta}(x+\delta)),y) \right]
> > > \leq \frac{1}{m} \sum_{i=1}^m \underset{\Vert \delta\Vert<\epsilon}{\mathrm{max}}\, \ell(w(g_{\theta}(x_i+\delta)),y_i) + 2c \cdot \tilde{\mathcal{R}}_S(\ell_F ) + 3c\sqrt{\frac{\log(2/\delta)}{2m}}, $$
> > >
> > > where  $\tilde{\mathcal{R}}\_S(\ell_F)$ is the Rademacher complexity, for which Theorem 3 in [1] provides bounds. In the linear probing setting, the bounds on the Rademacher complexity from Theorem 3 in [1] may include the norm of the data sample $g_{\theta}(x_i)$, $i=1,\dots,m$, instead of $x_i$. Does this align with your view on extracting a corollary for linear probing based on the results in [1]?
> > >
> > > In contrast to that, our result is focused on the difference between clean and robust risk. In particular, in Lemma 1, which directly follows from a crucial step in the proof of our Theorem 1, we bound the difference between the empirical clean and robust loss, i.e.,
> > > $$  \frac{1}{m} \sum_{i=1}^m  \ell(w(g_{\theta}(x_i)),y_i) - \frac{1}{m} \sum_{i=1}^m \underset{\Vert \delta\Vert_{} <\epsilon}{\mathrm{max}}\, \ell(w(g_{\theta}(x_i+\delta)),y_i). $$
> > > To the best of our understanding, the difference between these two risks is not considered in [1] and [2], and our bound can not be derived from their results.

---

> > > > ### Author Response · Authors · 2023-08-18
> > > >
> > > > Furthermore, bounding the difference between these two risks in terms of the robustness of the underlying representation function is a crucial characteristic of Theorem 1, given the goal of our work is to describe how the robustness of the representation function (which is often pretrained) contributes to the gap between the robust and clean risk (which is assessed w.r.t. downstream tasks). With that said, such a corollary from [1] could, however, be an opportunity to strengthen our results by replacing Hoeffding's inequality with a result derived from [1] in our proof. We will elaborate on the opportunity in a discussion in the revision, thanks again for bringing this to our attention.

---

> > > > ### Comment · Reviewer_MCBm · 2023-08-21
> > > >
> > > > Thank the authors for the response. I read the proof of Theorem 1 again and recognize that the second term is indeed not a complexity term, which implies that the results cannot be viewed as a direct corollary of [1] or [2]. This partially clarifies my confusion. However, I am still not convinced by their explanation. As is claimed in my review, I think that the main theoretical result (i.e., Theorems 1) is essentially a robust risk bound for linear classifiers. According to the rebuttal, I think the authors largely agree with my claim, despite their explanation that the results are “novel and provide a relevant contribution to studying robustness in transfer learning”.
> > > >
> > > > I agree that the idea is novel and the results bring new insights into the research of robustness in transfer learning. However, I think that a non-trivial deduction to the main theorem is necessary for a theoretical paper, especially when a large number of existing works have provided sophisticated similar results. The theoretical result seems somewhat trivial. Besides, this paper is far from 9-page long and takes up almost a whole page to discuss some "possible" future directions in Section 7. Although the length of the paper is not a reason for rejection, I doubt the amount of contribution of this paper based on the theoretical results. In my review, I recommend the authors accomplish the future work that is mentioned in Section 7, but the authors only provide possible changes in future revisions instead of making any substantial changes to the current paper.
> > > >
> > > > In summary, I think the theoretical contribution of the current version of this paper is not enough for this conference. My final decision is to maintain the current score.

---

> > > > > ### Author Response · Authors · 2023-08-21
> > > > >
> > > > > We would like to add a few clarifications regarding your response.
> > > > >
> > > > > 1) While we agree that Theorem 1 can be viewed as a bound on the robust risk of a linear predictor, as explained in [our prior response](https://openreview.net/forum?id=D8nAMRRCLS&noteId=9sMYPb2Cf5), the results are "novel and provide a relevant contribution to studying robustness in transfer learning” because bounding the robust risk of a linear predictor by the robustness of its underlying representation function provides a theoretical lens on the standard practice of transferring a pre-trained representation function to downstream tasks.
> > > > >
> > > > > 2) Regarding incorporating the future work from section 7 into the present submission, as stated in [our prior response](https://openreview.net/forum?id=D8nAMRRCLS&noteId=8eyzRl8QjT), the results and discussion in the W2 section of said response will be incorporated in our revision, which includes the experiments performed in response to your review shown in Table 1 in the added pdf, where, in accordance with our theory, reducing the norm of the linear predictor's weight matrix improves robustness transfer. To our knowledge, edits to the current paper were not allowed during the discussion period.

---

### Official Review · Reviewer_2ud3 · 2023-07-05

**Soundness:** 3 good
**Presentation:** 2 fair
**Contribution:** 2 fair
**Rating:** 6
**Confidence:** 3

**Summary:**

This paper study how the robustness properties of a given representation might be transferred to a linear probe which was trained on this representation.  Since the linear probe is a simple multiplication between the weights parameters and the representation vector, the authors were able to demonstrate that by using the representation we can bound the downstream linear probe robustness performances. To do so, they introduced a robustness score, which measure the maximal distance in the representation space that can be reached through adversarial perturbations. If this distance if lower than the linear classification margin, then we can expect the linear probe to be robust to these adversarial pertubations.

**Strengths:**

The paper is well written and easy to follow.

**Weaknesses:**

- There isn't a discussion about how regularizing the weight matrix of the linear probe might impact the downstream robustness. It's only one hypothesis but if within a representation vector, there might be some dimensions that are changing much more than some others, applying a sparse matrix multiplication might removed those dimensions and thus offer better robustness downstream.

- Some confusion about the name of robustness score. If this score is measuring how robust a representation is, then we could expect that a higher score will imply a better robustness while a lower score will offer weaker robustness.  However, in this instance, it doesn't seem to be the case, since a smaller robustness score imply better robustness. So I would advise the authors to rename their score of to change their formula such that higher robustness score imply better robustness.

- Some issues and limitations with the experimental setup. One hypothesis given by the authors for the results in Table 2 for which the proportion of fulfilled data points (condition 1) is zero, is that the resolution gap between the pretrained data and dowstream data is too big. However, since the authors are using a convnet for their experiment (wide-resnet), they should be able to change the resolution. So instead of resizing to 256x256, the authors could have only rescale to 64x64 (or 128x128) and compute their results from here. So I would have expected a better empirical evaluation of this discrepancy in the results. I also would have expected some discussion and study around the role of hyper-parameters when training the linear probe on condition 1. Lastly, the authors use adversarial pretrained model for their experiments, however, I would have expected to see some results on baseline trained without adversarial training.

- Inconsistency between some experiments. In Table 2, for the pretrained ImageNet model, the cifar10 accuracy is 92% while the robustness accuracy is 0%. However when looking at Table 8 the cifar10 accuracy is still 92% but the robustness accuracy is 48%. So if both table show linear probing accuracy, why is there those differences ?

**Questions:**

- In Table 2, you wrote that the proportion of fulfilled data points (condition 1) is zero for CIFAR10 and F-MNIST when pretrained on ImageNet. However, you wrote that the accuracy of fulfilled data points is 100%. I don't understand how the accuracy can be 100% if you have zero point in this category (If this is the case, it should be undefined, not 100%).
- Why the results for LP in Table 8 are different from Table 2 ?


**Limitations:**

The authors correctly addressed the limitations of their work.

---

> ### Author Rebuttal · Authors · 2023-08-09
>
> Thank you for your thorough feedback. In the following, we would like to comment on the weaknesses and questions raised in your review.
>
> **Weaknesses:**
> 1. Thank you for highlighting this topic. We agree that the impact of regularizing the weight matrix of the linear probe on downstream robustness should be discussed, and we will include it in the revision. According to our theoretical bound, the smaller the norm of the weight matrix, the more robustness transfer would be improved. For empirical affirmation, we provide new experimental results in Table 1 in the added pdf. Here, we repeat the experiments from Table 1 in our article, where CIFAR-100 is the pre-training set, but with weight matrix regularization during optimization (\lambda=0.01). The results indicate that regularization leads to a reduced weight norm and a reduced difference between clean and robust loss, i.e., better robustness transfer. The robustness score is not affected by these changes, as it only depends on the pre-trained feature representation of the downstream dataset. Interestingly, the left-hand-side (LHS) of our Lemma 1 bound remains relatively stable as well.
> 2. Thanks for highlighting this. Based on your feedback we will rename the score in the revised paper to “adversarial sensitivity score” (AS score) to make the name more in line with the concept. For this rebuttal, however, we will continue using the name robustness score to avoid confusion.
> 3. As you have observed correctly, we suggested the resolution gap between the datasets, e.g., ImageNet and CIFAR-10, as a possible reason for the poor robustness transfer but did not study it empirically. We have conducted such an analysis by considering different input image sizes, which are provided in Table 4 of the added pdf. The results show that the robustness transfer improves (smaller LHS and RHS of Lemma 1) if the input images are only resized to 64x64 instead of 256x256. At the same time, the clean performance drops when lowering the input resolution.
> Regarding hyperparameters for linear probing, we tuned on the validation loss, where the considered loss function is the same as used during training, i.e., MSE or CE loss. To clarify, the linear probe was trained on the corresponding loss using standard training (no adversarial examples involved), and condition (1) is not involved during training. The point of condition (1) is to evaluate not just if the predictions on our test examples are robust but if they are sufficiently separated to fulfill the theoretical bound. If we missed the point of your comment, it would be great if you could clarify what you meant.
> Regarding a baseline trained without adversarial training, in Appendix B.2, we consider such a baseline comparison. We perform transfer learning for a regression task and compute LHS and RHS of our theorem for pre-trained models, which are obtained from standard training, adversarial training, or random initialization.
> 4. The difference you have observed between the robust accuracy scores in Table 2 and Table 8, is due to different attack parameters in the two experiment setups. In both cases, we computed the adversarial examples using the Linf-PGD attack implemented in the foolbox package with attack strength ϵ = 8/255 and 20 steps. However, the relative step sizes vary between the two tables, 1/30 was used in Appendix B.4, while 0.7 was used in Section 5.2. This is because, for the experiments in Section 5.2, we performed a hyperparameter search on the attack parameters, but were not able to re-run the experiments in Appendix B.4 again. This is briefly mentioned in Appendix B.4.
> For the revision, we will re-evaluate B.4 to provide consistent robustness results throughout the experiments. With that said, we still want to note that the conclusions drawn from the experiments are still valid.
>
> **Questions:**
> 1. Thank you for bringing this to our attention, this is indeed a mistake on our side and we will correct it in the final version of the paper.
> 2. Please refer to our response to weakness 4.

---

> > ### Comment · Reviewer_2ud3 · 2023-08-14
> >
> > Thank you for your answers ! I agree that the “adversarial sensitivity score” is a much better name which will add a lot of clarity in the paper.  Looking at the different answers and other reviews, I decided to raised my score. Even if the first version of the paper might not satisfy the criteria for acceptance, I believe that the authors should be able to improve significantly their next revision to make it worth for acceptance.

---

### Official Review · Reviewer_dV4u · 2023-07-06

**Soundness:** 3 good
**Presentation:** 3 good
**Contribution:** 3 good
**Rating:** 5
**Confidence:** 3

**Summary:**

The paper studies the robust transfer of a pre-trained model, which demonstrate that the robustness of a linear predictor on downstream tasks can be constrained by the robustness of its underlying representation, and theoretically prove a bound on the difference between clean and adversarial loss in terms of the robustness score. The experiment further demonstrates the effectiveness of the ‘robustness score proposed’ in the paper.

**Strengths:**

1. The paper is well-written and the proposed theories are clearly formulated.

2. The paper provides theoretical guarantees for the robust transfer of pre-trained models.

3.  The experiment further demonstrates the effectiveness of the proposed theories.

**Weaknesses:**

See questions

**Questions:**

1 In Figure 1, How do define and distinguish between robust representation and non-robust representation?

2. The experiments in the paper are all transferred from pre-trained models on larger-scale data to small-scale data, whether it is also possible to transfer from small-scale to large-scale data.

3. How does the robustness of a pre-trained model affect the performance of the models on downstream tasks?


**Limitations:**

The pre-trained model is used as model initialization for different tasks. In the paper, it is only demonstrated on the classification task. It is not clear whether the proposed theory and method can be used for other tasks (e.g., object detection).

---

> ### Author Rebuttal · Authors · 2023-08-09
>
> Thank you for your review. In the following, we address all your questions and concerns.
>
> **Questions:**
> 1. In Figure 1, we distinguish between robust and non-robust representations based on their response to adversarial perturbations. We consider two separate representation functions in green and red. Both functions can map the same input image to slightly different feature representations. The robust representation (red) is less affected by adversarial perturbations, meaning that the model's feature representation remains stable and close to the representation of the original image even in the presence of small perturbations to the input data. On the other hand, the non-robust representation (green) results in significant changes to the model's representation vector when subjected to adversarial perturbations, indicating vulnerability to such attacks. This difference in adversarial sensitivity between robust and non-robust representation functions affects how robust, or non-robust, a linear classifier trained upon this feature representation space is, which is the focus of our study.
> 2. While we indeed only considered the large-scale to small-scale case, as this is the standard scenario where transfer learning is applied, our theory still applies to the small-scale to large-scale case. To validate this empirically, we provide additional results in Table 2 in the added pdf. Here, we study transferring from CIFAR-10 (small-scale dataset) to CIFAR-100 (a large-scale dataset). The results again confirm our theoretical bound.
> 3. While we focused on robustness transfer in particular, we agree that the link between pre-trained robustness and performance is a relevant question as well. In Table 3, we can see that there can be a trade-off between robust and clean performance by comparing the downstream performance on the CIFAR-10 and FashionMNIST tasks for the two different pre-trained models. In Table 3 in the extra pdf, we provide an additional experiment comparing robustness transfer and performance depending on two adversarial pre-training schemes. From these results, one can also conclude that in some cases the robustness can also be improved without sacrificing performance. In all, our experiments suggest directions to explore, and we will highlight said directions in the revision.
>
>
> **Limitations:**\
> In AppendixB.2, we provide experiments similar to Section 5.1 but considering a regression task and the MSE as the loss function.
> Further, we describe various loss functions for which our theory holds in Section 3.1. Thus, for all tasks which are fine-tuned using one of these loss functions our bound holds. Object detection is, indeed, an interesting special case as well. If considering a loss function that is a weighted sum of CE loss for the classification task and MSE for the localization task, given our empirical validation for classification and regression, our theory is also applicable to object detection applications.

---

> > ### Comment · Reviewer_dV4u · 2023-08-16
> > **Reply to the author**
> >
> > Thank you for the authors' response. After reading the rebuttals and other reviewers' comments, I keep my rating

---

### Official Review · Reviewer_9x4B · 2023-07-27

**Soundness:** 3 good
**Presentation:** 3 good
**Contribution:** 3 good
**Rating:** 6
**Confidence:** 4

**Summary:**

The paper considers the problem of robustness transfer from pretraining to downstream finetuning. The authors focus on the setup of linear probing, where downstream tasks are learned with a linear model on top of frozen pretrained features. They however give some insights on how to go beyond linear probing and what happens when the model is fully finetuned.

The paper is composed of a theory part, followed by an empirical analysis to derive practical insights from the theory. In the first part, the authors prove a bound on the difference between the adversarial and clean loss that is independent of the downstream task. The only conditions required for the this bound to hold are the Lipschitz continuity of the loss with respect to its first argument and the loss boundedness. The authors further highlight a number of loss functions that are valid in the scope of their theory, including the softmax cross entropy loss and the MSE. The authors further provide a criterion for robust classification, providing a minimum margin value that is sufficient for robustness. A notable aspect of their theory is its independence on the pretraining approach.

In the second part, the authors provide insights on how to calibrate expectation of robustness for downstream tasks. They validate their results with models pretrained on CIFAR100 and ImageNet, adversarially trained with softmax cross entropy and $L_\inf$ attacks, and consider CIFAR10, FashionMNIST and Intel Image as downstream tasks.

**Strengths:**

* The paper is clearly written and well structured. The related works are properly addressed and the papers brings interesting novel insights.
* The theoretical results are sound and make sense. The empirical analysis is well designed to validate the theory. (Both could be improved for higher impact, see next sections).
* The contributions are significant and important to the community, especially in the context of the increasing interest in the pretraining -> fine tuning paradigm.
* The theory is general enough, and applies in interesting practical settings.

**Weaknesses:**

* While the theory is general and applies to different objectives and pretraining approaches, the empirical analysis seems narrow, focusing on image classification with adversarial pretraining. It would be interesting to show how the results generalise to other modalities, and in case where robustness upstream doesn't come from explicit guards against attacks, but from other properties, e.g. scale.

* While the theoretical bound is interesting, there seems to be an opportunity to make it tighter with more (but still reasonable) conditions. Instead of the Hoeffding’s inequality, one can consider other more precise inequalities.

**Questions:**

* Most of the settings explored in the paper consider fairly similar pretraining and downstream tasks. What happens when the distribution shift is more severe? (The author did mention this is a interesting direction for further research, but empirical insights could make the paper stronger, even without further theory at this point).

* What happens when pretraining is done without explicit guard against adversarial attacks?

* In table 3, wouldn't it be more interesting to consider correlation between the robustness score and the clean & robust cross-entropy loss instead of their relative difference? For more data points, one could consider different checkpoints during training, that will give different robustness levels.

**Limitations:**

The authors did consider how they can improve their work and further developments, without explicitly stating the limitations. This seems reasonable to me given the nature of the work.

---

> ### Author Rebuttal · Authors · 2023-08-09
>
> Thanks for your review and feedback. We would like to comment on the weaknesses and questions raised in your review.
>
> **Weaknesses:**
> 1. While we agree that empirical validation in additional modalities and with large-scale models would be valuable, we have limited computational resources. With that said, we'd like to emphasize that the theory holds regardless of the source of the pre-trained model's robustness. Nonetheless, we’ll add these experiments as directions for future work in our revision to encourage further investigation by the community. In addition, we’d like to draw your attention to Appendix B.2, where we consider a regression task and evaluate our theory on models pre-trained using standard and adversarial training, as well as randomly initialized models. The results in Table 5 confirm that the theory holds regardless of the pre-training approach, and we also see that the robustness score (RHS) is expectedly much smaller if the model is adversarially trained.
> 2. A more precise inequality that we are aware of is Chernoff’s bound. This bound, however, requires the moment-generating function of the considered random variables and thus requires additional information/assumptions on the distribution of the difference between clean and adversarial loss. Given Hoeffding's inequality is valid regardless of the underlying distribution, we chose to use it for our proof. Please let us know if you had another inequality in mine that would be applicable.
> Finally, it’s worth noting that Hoeffding’s inequality is not the most critical factor prohibiting a tighter bound on the empirical robustness, note that the finite-sample version of Theorem 1 (Lemma 1), which was used for empirical validation, does not use Hoeffding’s inequality. What’s more critical is the Lipschitzness of the loss function and the bounds necessary for separating the effect of the linear probe’s weight matrix from the representation function.
>
> **Questions:**
> 1. Thank you very much for bringing this to our attention. We agree that the distribution shift between pre-training and downstream tasks should be discussed in the paper in more detail and we will do so in the revised version. In our experiments, we already consider scenarios with a severe distribution shift, e.g., ImageNet$\rightarrow$F-MNIST. In Table 3, we can observe that, as a result of the distribution shift, the robustness score increases.  We will make it clear that conducting a theoretical analysis to explain these empirical findings remains open to future work.
> Additionally, for the revision, we will incorporate a more granular analysis of the effect of distribution shift by performing transfer learning from ImageNet to corrupted ImageNet (ImageNet-C) with varying corruption strengths to obtain a fine-grained measure of the impact distribution shift has on robustness score.
> 2. A comparison of our bound with adversarially pre-trained, cleanly pre-trained, and randomly initialized networks is provided in Appendix B.2 in Table 5. The results confirm that the theory is valid independent of the pre-training approach and does not require the pre-trained model to be robust.
> 3. Regarding Table 3, while we do provide clean and robust CE loss, we decided to focus on relative difference, since it allows us to compare between tasks with different loss magnitudes, e.g., ImageNet vs Intel Image. Note that our theory considers the absolute difference between clean and robust loss, but relative to the norm of the linear probe’s weight matrix.
> Regarding different checkpoints, while we do not have pre-training checkpoints available, we nevertheless had a closer look at how robustness changes during transfer learning. In Figure 1 in the added pdf, we can see that the norm of the weight matrix (here the maximum of row-wise L2 norms) and the difference between clean and robust CE loss increased during transfer learning, while the LHS of our bound from Lemma 1 remains quite stable, as the numerator and denominator increase almost proportionally. Regarding the RHS, the robustness score remains constant, as it only depends on the pre-trained model processing the downstream dataset.

---

> > ### Comment · Reviewer_9x4B · 2023-08-16
> > **Post-rebuttal comment**
> >
> > I thank the authors for their answers. Most of my concerns were addressed. After considering the other reviews and the authors answers, I decided to raise my score. I think the paper can be improved, but it would be interesting to the community and can be built upon in future works.

---

### Author Rebuttal · Authors · 2023-08-09

We want to thank the reviewers for their thorough reviews and meaningful suggestions which will help us improve our paper.
We are also happy to hear that our article is considered “*well written and easy to follow*” (R-*2ud3*), that our contribution is “*significant and important to the community*” (R-*9x4B*) and the experiments demonstrate “*the effectiveness of the proposed theories*” (R-*dV4u*) and it provides an applicable “*robustness score, which is easy to calculate and is useful in estimating the robustness of the downstream classifiers*” (R-*MCBm*).

In the following, we state the primary takeaways from the reviews that will be incorporated into the revision:
- We would like to highlight here that we did also consider pretrained models that were not adversarially trained and scenarios different from classification. The corresponding regression experiments can be found in Appendix B.2.
- As briefly mentioned in Section 5.2, our experiments do consider the relationship between distribution shift and robustness transfer. A more detailed discussion on this topic with respect to the robustness score is relevant, as highlighted by reviewer *9x4B*, and will be added to the revision. This new discussion will include a more granular analysis of the effect of distribution shift by performing transfer learning from ImageNet to corrupted ImageNet (ImageNet-C) with varying corruption strengths.
- Following the suggestion by reviewer *2ud3*, we will add a separate discussion about how our theory provides insights into the effectiveness of weight norm regularization on robustness transfer. The empirical study in Table 1 of the added pdf confirms the theoretical statement that a smaller weight norm improves robustness transfer.

---

### Decision · Program_Chairs · 2023-09-21

**Decision:**

Accept (poster)

**Comment:**

The recommendation is based on the reviewers' comments, the area chair's personal evaluation, and the post-rebuttal discussion.

This paper studies the transferability of adversarial robustness via linear probing on pretrained models, which is a timely and important topic. All reviewers find the studied setting novel and the results provide new insights. The authors’ rebuttal has successfully addressed the major concerns of reviewers.

During the discussion, one reviewer pointed out that if this submission is positioned as a theory-oriented paper, the technical depth may need to be further sharpened; while if positioned as an experiment-driven paper, the empirical results may need to be expanded beyond supporting the theoretical statements. However, the reviewer also agrees that this submission has provided some new insights. Other reviewers are in favor of accepting this submission. Therefore, considering the joint contributions to new theoretical analysis and experimental results are sufficiently significant, I recommend acceptance of this submission. I also expect the authors to include the new results and suggested changes during the rebuttal phase to the final version.